# Polyphenol Profile, Antioxidant Activity, and Hypolipidemic Effect of Longan Byproducts

**DOI:** 10.3390/molecules28052083

**Published:** 2023-02-23

**Authors:** Si Tan, Zunli Ke, Chongbing Zhou, Yuping Luo, Xiaobo Ding, Gangjun Luo, Wenfeng Li, Shengyou Shi

**Affiliations:** 1School of Advanced Agriculture and Bioengineering, Yangtze Normal University, Chongqing 408100, China; 2South Subtropical Crops Research Institute, Chinese Academy of Tropical Agricultural Sciences, Zhanjiang 524091, China; 3Basic Medical School, Guizhou University of Traditional Chinese Medicine, Guiyang 550025, China; 4Luzhou Academy of Agricultural Sciences, Luzhou 646000, China

**Keywords:** longan, polyphenols, lipid metabolism, PPARα, LXRα

## Abstract

Longan, a popular fruit in Asia, has been used in traditional Chinese medicine to treat several diseases for centuries. Recent studies have indicated that longan byproducts are rich in polyphenols. The aim of this study was to analyze the phenolic composition of longan byproduct polyphenol extracts (LPPE), evaluate their antioxidant activity in vitro, and investigate their regulating effect on lipid metabolism in vivo. The results indicated that the antioxidant activity of LPPE was 231.350 ± 21.640, 252.380 ± 31.150, and 558.220 ± 59.810 (mg Vc/g) as determined by DPPH, ABTS, and FRAP, respectively. UPLC-QqQ-MS/MS analysis indicated that the main compounds in LPPE were gallic acid, proanthocyanidin, epicatechin, and phlorizin. LPPE supplementation prevented the body weight gain and decreased serum and liver lipids in high-fat diet-induced-obese mice. Furthermore, RT-PCR and Western blot analysis indicated that LPPE upregulated the expression of PPARα and LXRα and then regulated their target genes, including *FAS*, *CYP7A1*, and *CYP27A1*, which are involved in lipid homeostasis. Taken together, this study supports the concept that LPPE can be used as a dietary supplement in regulating lipid metabolism.

## 1. Introduction

Longan (*Dimocarpus longan* Lour.), also known as guiyuan, belongs to the family of *Sapindaceae*. It is widely distributed in Southeast Asia and Southern China, especially in Guangdong, Guangxi, and Fujian, and China accounts for 60% of the world’s longan production [1]. Longan has become one of the most popular fruits worldwide due to its delicious flavor and nutritional values. Longan fruits are rich in carbohydrates, fiber, vitamin C, amino acids, and so on. In China, longan is considered to be a “Medical Food Homology” and has been used in traditional Chinese medicine to cure insomnia and neural pain and to stop sweating and bleeding [2]. Longan can be consumed in both fresh and processed products such as ointment, wine, and canned longan. Normally, the pericarps and seeds, accounting for about 30% of the whole fresh fruit, are usually discarded as waste or burned after processing, which not only pollutes the environment but also wastes exploitable resources [3]. Recently, numerous studies have indicated that longan fruits are abundant in phenolic compounds and have strong antioxidant activity [4]. In particular, the types and contents of phenolics in the pericarp and seeds of longan are much richer than those in the pulps [5]. Additionally, polyphenolic extracts from longan pericarps or seeds have been reported to possess a variety of biological activities such as the promotion of healing of deep second-degree burns in mice [6], anti-hyperglycemic activity [2], anti-tyrosinase activity [7], and so on. However, to our knowledge, there is limited information regarding the phenolic profile of longan byproducts, and their effect on lipid metabolism has not been investigated.

Abnormal lipid metabolism is an important risk factor of obesity, hyperlipidemia, fatty liver, and type 2 diabetes, as well as other chronic metabolic diseases. Several mechanisms, including regulation of cholesterol absorption, inhibition of synthesis and secretion of triglyceride, and reduction in low-density lipoprotein oxidation in plasma, contribute to maintaining or improving lipid metabolism [8]. Various factors could influence lipid metabolism, such as eating habits, exercise, and gut microbiota. Growing evidence suggests that many polyphenols are involved in lipid metabolism and, therefore, may prevent obesity [9]. For example, Goishi tea polyphenols might reduce cardiovascular disease risk by lowering triglyceride levels, according to a randomized, double-blind, placebo-controlled clinical study [10]. In addition, citrus polyphenols have been reported to regulate lipid metabolism and prevent metabolic diseases, according to numerous animal and human clinical trials [11]. In all, polyphenol-rich dietary supplementation may improve lipid metabolism and prevent metabolic syndrome.

Multiple pathways are involved in lipid metabolism, and peroxisome proliferator-activated receptors (PPARs) play an important regulatory role. In particular, PPARα is highly expressed in the liver and controls a variety of metabolic processes in the liver, including mitochondrial fatty acid oxidation, fatty acid binding, degradation of triglyceride, lipid synthesis, and so on [12]. In addition, liver X receptor α (LXRα) is an important nuclear receptor transcription factor, the expression of which is highly correlated with hepatic adipose deposition and plays a key role in the transcriptional regulation of cholesterol homeostasis [13]. PPARα and LXRα interact to influence the regulation of fatty acid and cholesterol metabolism. For example, by regulating PPARα and LXRα pathways, sesamin ameliorates hepatic steatosis induced in rats fed a high-fat diet [14]. Moreover, Ge et al. also reported that, by regulating PPARα and LXRα pathways, betaine prevented nonalcoholic fatty liver diseases induced by fructose in rats [15]. The important role of PPARα and LXRα in lipid metabolism renders them important targets for pharmacological and dietary approaches to improve lipid metabolism, obesity, and metabolic syndrome.

The aim of this study was to analyze the phenolic composition of longan byproduct polyphenol extracts (LPPE), evaluate its antioxidant activity in vitro, and investigate its regulating effect on lipid metabolism in vivo. First, the polyphenols of longan byproducts were extracted, and the chemical profile was analyzed via high-pressure liquid chromatography, and triple quadrupole mass spectrometry (UPLC-QqQ-MS/MS). Then, the ability of LPPE to lower blood and liver lipids, thereby improving hepatic steatoses, was evaluated. Furthermore, quantitative real-time PCR (RT-PCR) and Western blot were carried out to reveal the molecular mechanisms.

## 2. Results

### 2.1. Polyphenol Contents and Antioxidant Activity of LPPE

The UPLC fingerprint of LPPE is shown in Figure 1. Eleven compounds including gallic acid, proanthocyanidin B2, epicatechin, proanthocyanidin A2, syringic acid, p-hydroxybenzoic acid, poncirin, ferulic acid, rutin, phlorizin, and methyl hesperidin were unambiguously identified. As shown in Table 1, the quantitative analyses of LPPE indicated that the principal component in LPPE was phlorizin (38.894 ± 3.765 mg/g), followed by proanthocyanidin A2 (24.382 ± 2.859 mg/g), gallic acid (24.080 ± 2.791 mg/g), and epicatechin (7.592 ± 0.231 mg/g).

The extraction rate of LPPE was 10.69%, and the total phenolic content of LPPE measured via the Folin–Ciocalteu colorimetric method was 285.350 ± 36.430 mg GAE/g. To understand the antioxidant activity of LPPE, its scavenging DPPH and ABTS radical abilities and ferric-ion-reducing antioxidant power (FRAP) were evaluated. As shown in Table 1, LPPE exhibited a good capacity for scavenging DPPH (231.350 ± 21.640 mg Vc/g) and ABTS (252.380 ± 31.150 mg Vc/g) radicals and FRAP (558.220 ± 59.810 mg Vc/g).

### 2.2. Effects of LPPE on Body Weight and Cell Size of Epididymal Adipose Tissues in High-Fat Diet-Induced Obese Mice

In this study, C57BL/6J mice fed with an HF diet were used as the lipid metabolism disorder model. As shown in Figure 2A, an HF diet led to a more than 60% higher body weight compared with the mean body weight in the Chow group after 12 consecutive weeks of feeding, which is a significant increase. Moreover, LPPE supplementation significantly decreased the high-fat-diet-induced body-weight gain and caused a mean 11.8% reduction in total body weight compared with the HF group, without affecting the food intake. An HF diet also led to a significant increase in the cell size of epididymal white adipose tissues, and 0.2% LPPE exerted an antagonizing effect (Figure 2B). Those results indicate that LPPE alleviated the mice obesity that was induced by an HF diet.

### 2.3. Effects of LPPE on Serum and Liver Lipids, and Hepatic Steatosis in High-Fat Diet-Induced Obese Mice

Long-term high-fat diet can easily lead to hyperlipidemia. As shown in Figure 3A, the mice in the HFD group exhibited significantly higher levels of serum TC, TG, HDL-c, and LDL-c when compared to those in the Chow group. Interestingly, significant reductions in serum TC, TG, HDL-c, and LDL-c levels were observed in 0.2% LPPE-treated mice. The results of the total liver lipid levels indicated that the highest contents of TG (Figure 3B) and TC (Figure 3C) were found in the HFD group, which were greatly alleviated by 0.2% LPPE supplementation. The histological analysis showed that excessive ballooning degeneration (Figure 3D) and lipid droplets (Figure 3E) occurred in the mice livers from the HFD group compared to those in the Chow group, suggesting that HFD induced hepatic steatosis. Expectedly, liver tissues in the LPPE group showed much less ballooning degeneration and lipid droplets, suggesting that hepatic fat accumulation and hepatic steatosis induced by a high-fat diet were strongly ameliorated by LPPE treatment.

### 2.4. Effects of LPPE on the Gene Expression Involved in Lipid Metabolism

To understand the underlying molecular mechanisms by which LPPE improve lipid metabolism, the expressions of genes associated with lipid metabolism were determined. In this study, two nuclear receptors, peroxisome-proliferation-activated receptor alpha (PPARα) and liver X receptor alpha (LXRα), which play important roles in lipid homeostasis, were analyzed [12,13]. A high-fat diet led to significantly decreased PPARα (Figure 4A) and LXRα (Figure 4B) gene expression. As expected, LPPE significantly enhanced the expressions of PPARα and LXRα. Additionally, fatty acid synthase (*FAS*) gene expression was significantly increased in the HFD group when compared to that in the Chow group, and LPPE inhibited the expression of *FAS* (Figure 4C). On the other hand, there were significant decreases in the expressions of the cholesterol 7-alpha hydroxylase (*CYP7A1*) (Figure 4D) and cholesterol 27-hydroxylase (*CYP27A1*) (Figure 4E) genes in the HFD group, which were then effectively up-regulated by LPPE supplementation.

### 2.5. Effects of LPPE on the Protein Expressions Involved in Lipid Metabolism

Western blot analysis (Figure 5 and Appendix A) revealed that the protein expression levels of PPARα, LXRα, and CYP7A1 were also inhibited by a high-fat diet, and 0.2% LPPE supplementation significantly elevated the expression of PPARα, LXRα, and CYP7A1. Similar to the gene expression result, LPPE also suppressed the expression of FAS, which was activated by a high-fat diet. In summary, the gene and protein analysis results indicated that PPARα/LXRα signaling is likely to contribute to the improvement of lipid metabolism mediated with LPPE.

## 3. Discussion

Lipid metabolism disorder is currently considered to be a hallmark characteristic of many chronic metabolic diseases, such as obesity and cardiovascular disease [16]. As is well known, dietary ingredients play key role in the development, progression, and prevention of lipid accumulation. For example, a high-fat diet induces obesity, whereas large quantities of phytochemicals such as polyphenols have shown potential regulating effects on lipid metabolism, including decreasing lipid accumulation in both the liver and the blood [8]. Therefore, it is promising to regulate lipid metabolism by dietary intervention.

Recently, phytotherapy, which is defined as the therapeutic use of whole or minimally modified plant components, has been focused on preventing diseases and improving health conditions [17]. In particular, polyphenols, which have shown strong antioxidant activity, have been reported to be effective in improving lipid metabolism in the literature [18]. For example, curcumin, a natural polyphenol from turmeric, has been reported to improve features of fatty liver according to a randomized, placebo-controlled trial [19]. In addition, blueberry polyphenols have been shown to inhibit body weight gain and return lipid metabolism to normal in high-fat diet-induced-obese mice [20]. Moreover, polyphenols in pomegranate peel alleviated inflammation and hypercholesterolaemia in high-fat diet-induced-obese mice [21]. In all, polyphenols and polyphenol-rich dietary supplementation are important approaches in improving lipid metabolism.

In this study, the polyphenol profile and antioxidant activities of longan pericarp extracts (LPPE) were analyzed. In total, 11 phenolic compounds were identified in this study, and most of those compounds in longan fruits have already been reported by several studies [1,4,22]. Our results indicated that LPPE were rich in phlorizin, proanthocyanins, gallic acid, and epicatechin. Phlorizin, a natural dihydrochalcone possessing several pharmacological activities such as antioxidant, anti-inflammatory, antidiabetic, and hepatoprotective activities [23], has been reported as one of the main flavonoid glycosides in the seeds of lychee [24]. Furthermore, Li et al. also reported that the longan pericarp extracts are rich in proanthocyanidin A, gallic acid, and epicatechin [2]. However, corilagin, which has been reported as one of the main phenolic compounds in longan by several studies [4,25], was not identified in this study. Similarly, this compound was also not detected in other studies [1,26,27]. This difference may result from the different cultivar, cultivation environment, purification method, and so on.

The total polyphenol content of LPPE was much higher than that reported by Prasad et al. [28], who indicated that the total polyphenol content of longan fruit pericarp determined via ultra-high-pressure-assisted extraction was about 100 mg GAE/g, but it was lower than the longan pericarp extracts purified via a Sephadex LH-20 column [2]. Those difference may be caused by the different cultivar, degrees of ripeness, and extraction and purification methods. In addition, the result of antioxidant activity of LPPE was similar to that reported by He et al. [27] when the extraction rate was considered, and it was much higher than most of medical plant samples such as ginger, ginkgo, and *Psidium guajava* [29,30]. All of those results indicated that LPPE is rich in the variety of phenolic compounds and has good antioxidant activity.

Furthermore, the effect of LPPE on lipid metabolism was firstly evaluated, and the results revealed that LPPE could decrease body weight and serum lipids and inhibit hepatic lipid accumulation induced by a high-fat diet. According to our component analysis, the main phenolic compounds in LPPE include gallic acid, epicatechin, proanthocyanidins, and phlorizin. Gallic acid showed a hypolipidemic effect in mice fed a high-fat diet [31] and protected the liver against nonalcoholic fatty liver disease by inhibiting inflammatory signaling pathways in Wistar rats [32]. Epicatechin, a natural flavanol which is found in green tea and cocoa, has also been reported to decrease blood lipids and attenuate hepatic steatosis in high-fat diet-induced-obese rats [33]. Furthermore, proanthocyanidins and proanthocyanidin-rich extracts have been observed to decrease serum lipids [34], and regulate lipid metabolism in rats at the molecular level [35]. Another dihydrochalcone phlorizin, which is mainly distributed in the plants of the *Malus* genus, has been reported to improve lipid metabolism in streptozotocin-induced diabetic rats [36] and to ameliorate lipid deposition in mice fed a high-fat diet [37]. Therefore, these phenolic constituents may contribute to the preventive effect of LPPE on lipid metabolism disorder. In addition, it is necessary to consider the synergistic effects of all the components of LPPE.

Earlier studies have shown that longan pericarps and seeds are rich in polyphenols and have strong antioxidant, anti-inflammatory, and anti-cancer activities [38]. In this study, our results firstly indicated that LPPE might be beneficial for improving lipid metabolism, partly through the PPARα and LXRα pathways. PPARα inhibits liver triglyceride synthesis by inhibiting the expression of *FAS* and other de novo fatty acid synthesis genes [12]. On the other side, PPARα agonists enhance fatty acid oxidation, increase lipid decomposition, and reverse cholesterol transportation [39]. Activation of LXRα can promote the expression of CYP7A1, which is an important rate-limiting enzyme in the process of bile acid synthesis by cholesterol, and thus plays a positive regulatory role in cholesterol metabolism [40]. Moreover, a loss of LXRα can lead to peripheral cholesterol accumulation and liver lipid deposition, promoting the progression of nonalcoholic fatty liver in mice fed a high-fat diet, whereas a high expression of LXRα can reduce liver inflammation and fibrosis induced by a high-fat diet [41]. In the present study, the transcriptional levels of *PPARα* and *LXRα* were increased in the LPPE group when compared to the HFD group, and their target genes *FAS*, *CYP7A1,* and *CYP27A1,* which are related to fatty acid and cholesterol metabolisms, were obviously regulated following supplementation with LPPE. Western blot results were also consistent with the gene expression results. Those results suggested that LPPE may regulate aspects of hepatic lipid metabolism, including lipid synthesis, lipid oxidation, and lipid transportation, thereby improving lipid metabolism disorders induced by a high-fat diet. Similarly, several dietary polyphenols or phenolic-rich extracts were reported to have hepatoprotective effects by regulating *PPARα* and *LXRα* pathways [42]. Interestingly, a previous study also showed that polyphenol-rich longan flower water extract had anti-obesity and hypolipidemic effects in rats fed a hypercaloric diet, upregulated *PPARα* gene expression, and decreased *FAS* gene expression. The main polyphenols in longan flower extract were gentisic acid, epicatechin, ferulic acid, and gallic acid, which had similar components to those in LPPE [43]. However, Liu et al. reported that longan flower water extracts attenuated nonalcoholic fatty liver by decreasing lipid peroxidation and downregulating matrix metalloproteinases-2 and -9 in rats [44]. The differences in the mechanisms may originate from the different animal models, chemical profile of extracts, and additive dose of polyphenols. Nevertheless, the above studies indicated that the mechanism by which polyphenolic compounds regulate lipid metabolism is complex. Taken together, we speculated that LPPE improved lipid metabolism partly through the PPARα/LXRα/FAS/CYP7A1 pathway.

## 4. Materials and Methods

### 4.1. Reagents and Standards

Acetonitrile (HPLC-grade) was purchased from Adamas reagent, Ltd. (Shanghai, China). All the polyphenol standards (gallic acid, proanthocyanidin B2, epicatechin, proanthocyanidin A2, syringic acid, p-hydroxybenzoic acid, poncirin, ferulic acid, rutin, phlorizin, and methyl hesperidin), purity ≥99%, were purchased from Shanghai yuanye biotechnology Co. Ltd. (Shanghai, China). Vitamin C; 1, 1-diphenyl-2-picrylhydrazyl (DPPH); 2, 2′-azino-bis (3-ethylbenzothiazoline-6-sulfonic acid) (ABTS); 2, 3, 5-triphenyltetrazolium chloride (TPTZ); triton; tris-HCl; paraformaldehyde; and urethane were purchased from the Beijing Solaibao Technology Co. Ltd. (Beijing, China). Trizol reagent was purchased from TaKaRa (Beijing, China). Hematoxylin and eosin (H&E) kit, Oil Red O (ORO) kit, and a protein extraction kit were obtained from Beyotime Biotechnology (Beijing, China). Antibodies were purchased from Abcam (Shanghai, China).

### 4.2. Polyphenol Extracting

Fresh longan fruits (Chuliang) at the commercial maturity stage were collected from the Institute of China Southern Subtropical Crop Research (Zhanjiang, China). The pericarps and the seeds were manually collected. After washing, those samples were frozen at −60 °C for 12 h. Then, the samples were dried in an experimental vacuum lyophilizer (LGJ-10, Shengchao kechuang biotechnology Co. Ltd., Beijing, China) for 72 h. The dried samples were ground with a pulverizer (FW-100, Beijing Ever Bright Medical Treatment Instrument Co., Ltd., Beijing, China) rapidly and sieved through a 60-mesh screen (screen size of 250 mm). The powder (1 kg) was ultrasonically extracted with an ultrasonic bath (KQ-5200B, Kunshan Ultrasonic Instrument Co., Ltd., Shanghai, China) at 40 °C for 30 min using a solvent of 80% aqueous ethanol (*w/v* = 1:5) [30]. The solid–liquid mixture was vacuum filtered, and the ethanol solution was concentrated under reduced pressure at 35 °C. Then, to remove the impurities and improve the purity of the polyphenols, the crude polyphenol extract was subsequently re-extracted using ethyl acetate (1:5, *v/v*). The ethyl acetate fraction was concentrated and then lyophilized. The longan byproduct polyphenol extract (LPPE) was stored hermetically at −20 °C for further analysis and animal studies.

### 4.3. Identification and Quantification of Polyphenols via UPLC-QqQ-MS

The polyphenol profile of LPPE was analyzed with an ultra-high-pressure liquid chromatograph (UPLC, Agilent, Santa Clara, CA, USA) and a triple quadrupole mass spectrometer (6460QqQ-MS, Agilent, Santa Clara, CA, USA) equipped with an electrospray ionization source (ESI). A ZORBAX Eclipse Plus C18 column (100 mm × 2.1 mm i.d. 1.8 µm, Agilent, Waldbronn, Germany) was used. The mobile phase consisted of 0.1% aqueous formic acid (A) and methanol with 0.1% formic acid (B) at a flow rate of 0.2 mL min^−1^. The gradient elution was set as follows: 0–10 min, 100% A–20% A; 10–11 min, 20% A; 11–12 min, 20%–100% A. The sample injection volume was 5 μL, and the column temperature was 35 °C. All the compounds were identified by comparing the dynamic multiple reaction monitoring (MRM) information with reference standards.

### 4.4. Total Polyphenols and Antioxidant Activities In Vitro

The total polyphenol content was analyzed via a modified Folin–Ciocalteu colorimetric method. Briefly, a 50 μL solution of extracts was mixed with 50 μL Folin–Ciocalteu Phenol reagent. After incubation in the dark for 6 min, 100 μL 7% Na_2_CO_3_ and 100 μL distilled water were added, and the mixture was incubated at room temperature for 90 min. The absorbance was measured at 760 nm via a microplate reader. The results were expressed in milligrams of gallic acid equivalent per gram of extracts (mg GAE/g). The antioxidant activities were determined with DPPH, ABTS, and FRAP assays. For the assay of DPPH, 20 μL LPPE solution was added into 100 μL of DPPH solution. After incubation at room temperature for 30 min, the absorbance was measured at 517 nm. For the assay of ABTS, 20 μL LPPE was added into 100 μL of ABTS solution (absorbance of 0.70 ± 0.02 at 734 nm). After incubation at room temperature for 10 min, the absorbance was measured at 734 nm. For the assay of FRAP, 20 μL LPPE solution was mixed with 100 μL FRAP solution (2.5 mL 20 mmol L^−1^ TPTZ, 2.5 mL 20 mmol/L FeCl_3_, and 25 mL 0.2 mol L^−1^ acetate buffer). The solution was incubated at 37 °C for 10 min, and absorbance was measured at 593 nm. Vc was used as the standard, and the antioxidant activity was expressed in mg Vc equivalents per gram of extracts (equivalent mg Vc/g) [30].

### 4.5. Animal Experiment

All the animal experiment protocols were approved by the Institutional Animal Care and Ethical Committee of Guizhou University of Traditional Chinese Medicine. Male wild-type C57BL/6J mice (6–8 weeks old, 22 ± 1 g) were obtained from Changsha Tianqin Biotechnology Co., Ltd. (Changsha, China). Mice were kept in a standard breeding room with temperature 22 ± 1 °C, humidity 60 ± 10%, and 12 h dark/light cycle. The mice could obtain food and water ad libitum and were allowed to acclimatize for 1 week before the experiment. Then, the mice were randomly divided into 3 groups (n = 8), fed with a Chow diet (10% of calories derived from fat, Research Diets, D12450B), or a high-fat diet (HFD) (60% of calories from fat, Research Diets, D12492), or a high-fat diet plus 0.2% LPPE for 12 weeks. The food intake and body weight were recorded once per week.

After 12 weeks of feeding, mice were fasted overnight (12 h) and then anesthetized with 20% urethane. Blood was collected via cardiac puncture, and the supernatant serum samples were separated via centrifuging at 1000× *g* for 20 min at 4 °C (centrifuge 5425, Eppendorf, Hamburg, Germany). Serum samples were stored at −80 °C for further analysis.

The liver and epididymal adipose tissues were collected individually. A small piece of liver or epididymal adipose tissue was fixed in 4% buffered formalin for histological analysis. The remaining livers were rapidly frozen in liquid nitrogen and then stored in −80 °C for hepatic lipid content, RT-PCR, and Western blot analysis.

### 4.6. Histological Analysis

The fixed sections of liver or epididymal adipose tissues were stained with hematoxylin and eosin (H&E). Another portion of the liver samples was frozen and embedded in OCT, and then sliced. Subsequently, the slices were stained with Oil Red O. The images were captured with a Zeiss Axio Imager microscope (Axio Imager M1, Zeiss, Oberkochen, Germany).

### 4.7. Serum Lipids

The serum triglycerides (TG), total cholesterol (TC), high-density lipoprotein cholesterol (HDL-c), and low-density lipoprotein cholesterol (LDL-c) levels were detected via biochemistry kits (Nanjing Jiancheng Bioengineering Institute, Nanjing, China).

### 4.8. Liver TG and TC Analysis

Liver lipids were determined as described before [45]. Briefly, about 50 mg of frozen liver tissues were homogenized in 0.5 mL lysis buffer and then mixed with an equal volume of chloroform. The chloroform layer was collected, dried overnight at room temperature, and resuspended in isopropyl alcohol. The hepatic TG and TC were detected according to the protocol of assay kits (Dongou Diagnostic Product Ltd. Wenzhou, China).

### 4.9. RT-PCR

Total RNA was extracted from liver tissues with the Trizol reagent. cDNA was synthesized using the cDNA synthesis kit (Thermo Scientific, Waltham, MA, USA). The gene expression levels were quantified with the ABI StepOne Plus real-time.

PCR system (Applied Biosystems, Foster City, CA, USA) with SYBR green.

Supermix (Bio-Rad, Hercules, CA, USA). The sequences of the primers used were as follows: *FAS* (forward primer, 5′CTGAGATCCCAGCACTTCTTGA3′; reverse primer, 5′GCCTCCGAAGCCAAATGAG3′); *PPARα* (forward primer, 5′AGGCTGTAAGGGCTTCTTTCG3′; reverse primer, 5′GGCATTTGTTCCGGTTCTTC3′); *LXRα* (forward primer, 5′TCAGAAGAACAGATCCGCTTG3′; reverse primer, 5′CGCCTGTTACACTGTTGCT3′); *CYP7A1* (forward primer, 5′AACAACCTGCCAGTACTAGATAGC3′; reverse primer, 5′GTGTAGAGTGAAGTCCTCCTTAGC′); *CYP27A1* (forward primer, 5′GCCTCACCTATGGGATCTTCA3′; reverse primer, 5′TCAAAGCCTGACGCAGATG3′). The mRNA expression in liver was calculated after normalization to *β-actin* (forward primer, 5′TGTCCACCTTCCAGCAGATGT3′; reverse primer, 5′AGCTCAGTAACAGTCCGCCTAGA3′).

### 4.10. Western Blot Assay

Hepatic protein expressions of FAS, PPARα, LXRα, and CYP7A1 were analyzed via the Western blot (WB) assay, as previously described [45]. Briefly, total proteins from liver cells were extracted and separated in SDS-PAGE and then transferred onto a PVDF membrane and blocked with 5% BSA for 2 h at room temperature. The membranes were incubated with antibodies (1:2500 dilution) overnight at 4 °C. Then, they were washed with TBST (10 min × 3) and incubated with horseradish peroxidase conjugated secondary antibody (1:2500 dilution) for 2 h at room temperature. The immunoreactive bands were shown, and the densitometry was quantified via the Gel Image Analysis System (Advansta, San Jose, CA, USA). The β-actin was used as the control protein.

### 4.11. Statistics Analysis

All the values are expressed as mean ± SEM. Statistically significant differences among the groups were analyzed via one-way analysis of variance (ANOVA), followed by Tukey’s post hoc test with SPSS (Version 15.0, SPSS Inc., Chicago, IL, USA). The significant difference level was set at *p <* 0.05.

## 5. Conclusions

In conclusion, LPPE was shown to be rich in phenolic compounds and high in antioxidant activity. Dietary supplementation of LPPE decreased body weight and serum lipids and improved hepatic steatosis in high-fat diet-induced-obese mice, suggesting it has a preventive effect on lipid accumulation. LPPE-improved lipid metabolism may occur partly through the PPARα/LXRα/FAS/CYP7A1 pathway. However, the main contributors need to be further investigated. This study provides an effective option in terms of a dietary supplement to improve lipid metabolism.

## Figures and Tables

**Figure 1 molecules-28-02083-f001:**
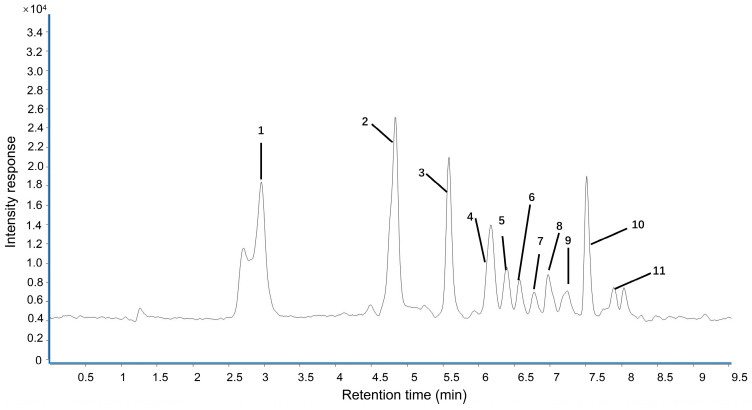
Representative chromatograms of the polyphenols in LPPE. 1—Gallic acid. 2—Proanthocyanidin B2. 3—Epicatechin. 4—Proanthocyanidin A2. 5—Syringic acid. 6—p-Hydroxybenzoic acid. 7—Poncirin. 8—Ferulic acid. 9—Rutin. 10—Phlorizin. 11—Methyl hesperidin.

**Figure 2 molecules-28-02083-f002:**
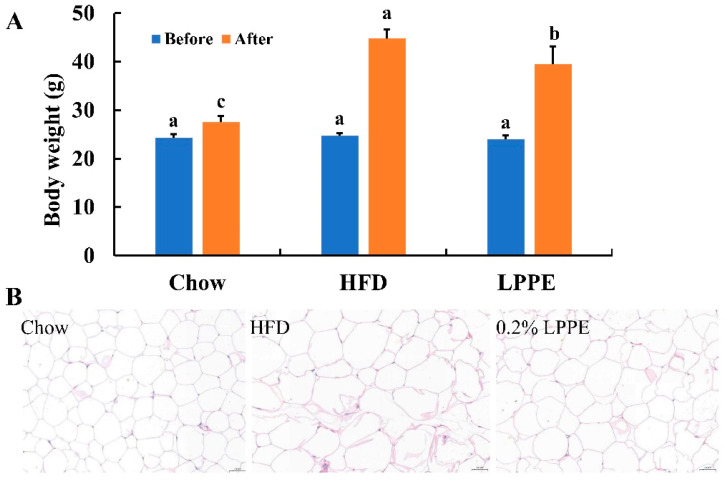
Effects of LPPE on the body weight and the cell size of epididymal adipose tissues in high-fat diet-induced-obese mice. (**A**) Body weight. (**B**) H&E staining (×200) of the epididymal adipose tissues. Values are expressed as the mean ± SEM (n = 8). Different letters above the bars with the same color indicate significant differences (*p* < 0.05).

**Figure 3 molecules-28-02083-f003:**
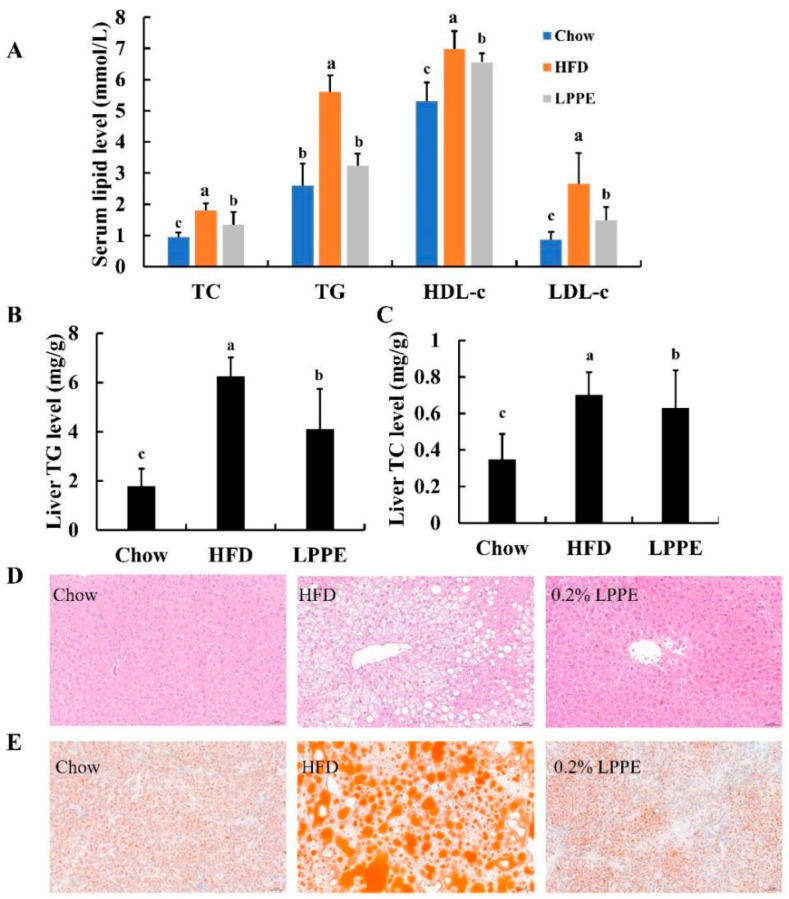
Effects of LPPE on serum and liver lipids, and hepatic steatosis in high-fat diet-induced-obese mice. (**A**) Serum TC, TG, HDL-c, and LDL-c. (**B**) Hepatic TG level. (**C**) Hepatic TC level. (**D**) H&E staining (×200) of hepatic tissues. (**E**) ORO staining (×200) of hepatic tissues. Values are expressed as the mean ± SEM (n = 8). Different letters above the bars with the same color index indicate significant differences (*p* < 0.05).

**Figure 4 molecules-28-02083-f004:**
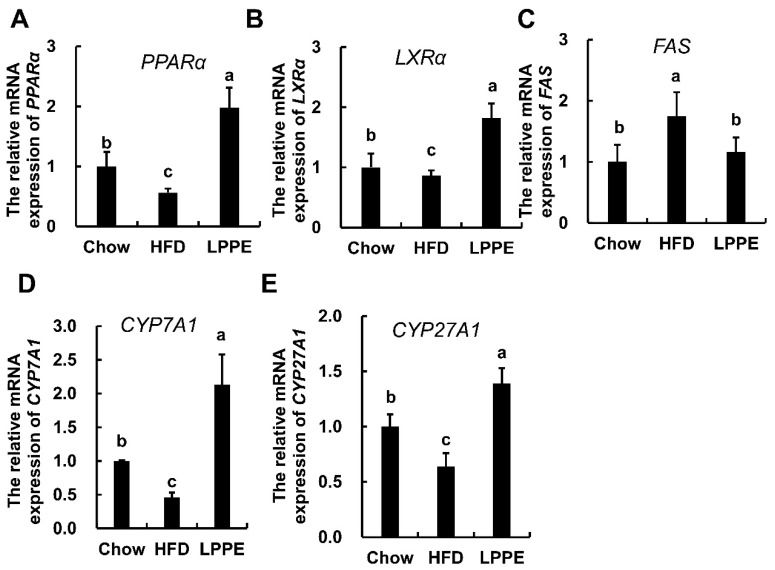
Effects of LPPE on the gene expressions involved in lipid metabolism. The mRNA expression of hepatic *PPARα* (**A**)*, LXRα* (**B**)*, FAS* (**C**)*, CYP7A1* (**D**), and *CYP27A1* (**E**). Values are expressed as mean ± SEM (n = 6). Different letters indicate significant differences (*p* < 0.05).

**Figure 5 molecules-28-02083-f005:**
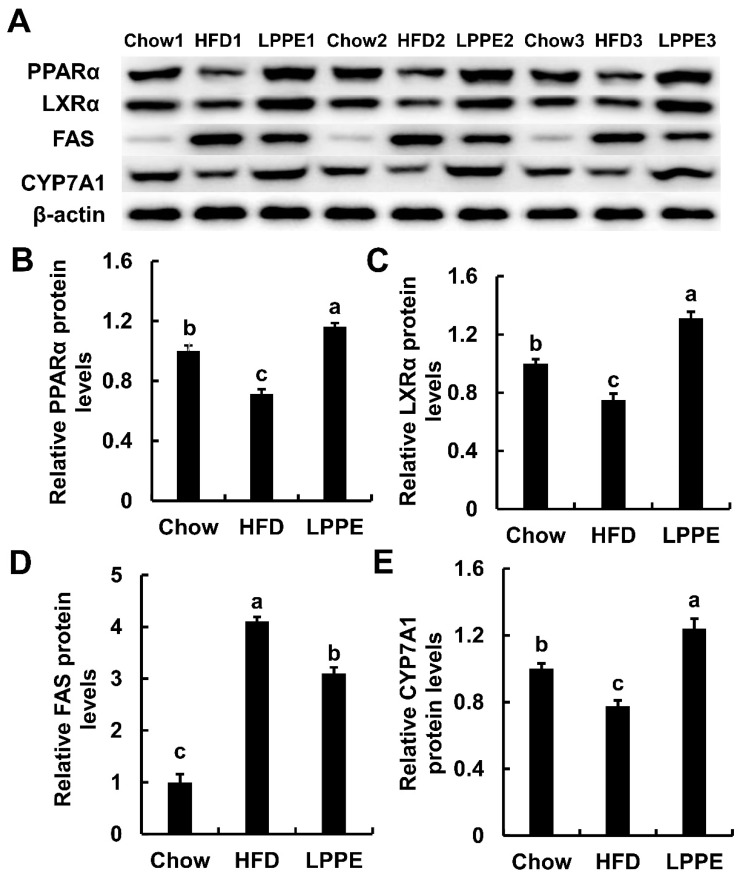
Effects of LPPE on the protein expressions involved in lipid metabolism. The Western blot photos (**A**). The relative PPARα protein levels (**B**). The relative LXRα protein levels (**C**). The relative FAS protein levels (**D**). The relative CYP7A1 protein levels (**E**). Values are expressed as mean ± SEM (n = 3). Different letters indicated significant differences (*p* < 0.05).

**Table 1 molecules-28-02083-t001:** Polyphenol composition, total polyphenols via Folin–Ciocalteu colorimetric method and antioxidant activities in vitro of LPPE. Values are expressed as the mean ± SEM.

No.	Compounds	Retention Time (min)	Fragmentor (V)	MS [M-H]^-^	MS/MS (*m*/*z*)	Contents (mg/g)
1	Gallic acid	2.966	100	168.9	125, 106.8	24.080 ± 2.791
2	Proanthocyanidin B2	4.838	100	577	289, 407	2.106 ± 0.245
3	Epicatechin	5.588	130	289	109, 203	7.592 ± 0.231
4	Proanthocyanidin A2	6.131	150	575	285, 423	24.382 ± 2.859
5	Syringic acid	6.426	100	197	122.8, 182	2.361 ± 0.106
6	p-hydroxybenzoic acid	6.584	130	137	137	0.279 ± 0.061
7	Poncirin	6.755	150	593	284.9	0.607 ± 0.044
8	Ferulic acid	6.984	90	192.9	134, 149	1.402 ± 0.077
9	Rutin	7.253	160	609	299.9	0.050 ± 0.004
10	Phlorizin	7.521	150	434.9	272.9, 167	38.894 ± 3.765
11	Methyl hesperidin	7.849	135	623	315, 338.7	0.488 ± 0.004
Total polyphenols					285.350 ± 36.430 (mg GAE/g)	
DPPH					231.350 ± 21.640 (mg Vc/g)	
ABTS					252.380 ± 31.150 (mg Vc /g)	
FRAP					558.220 ± 59.810 (mg Vc/g)	

Note: GAE—gallic acid equivalent; Vc—ascorbic acid.

## Data Availability

The authors declare that the data supporting the findings of this study are presented within the manuscript. Additional data sources are also available from the corresponding author upon reasonable request.

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
