# Peer review of "Polyphenol Profile, Antioxidant Activity, and Hypolipidemic Effect of Longan Byproducts"

_molecules, 2023, doi:10.3390/molecules28052083_

Round 1

Reviewer 1 Report

The current study by Tan et al., is an attempt to evaluate the phenolic composition of longan byproducts polyphenol extracts. Then determine its antioxidant and hypolipidemic effect. The work is valuable and contains important findings and I would like to recommend the publication, however the following issue need to be clarified:

1- In animal experiment, the researchers used mice, however it is usually better to use rats. Mice are too small to deal with.

2- Why you did not compare all the evaluated parameters between treated and untreated animals.

Author Response

The current study by Tan et al., is an attempt to evaluate the phenolic composition of longan byproducts polyphenol extracts. Then determine its antioxidant and hypolipidemic effect. The work is valuable and contains important findings and I would like to recommend the publication, however the following issue need to be clarified:

1- In animal experiment, the researchers used mice, however it is usually better to use rats. Mice are too small to deal with.

Answer: we appreciate reviewer’s good comments. As we know, both rats and mice were used as the animal models. In our lab, a complete set of sophisticated mice testing techniques has been developed. Therefore, mice were used in this study.

2- Why you did not compare all the evaluated parameters between treated and untreated animals.

Answer: all the evaluated parameters including body weight, serum lipid levels, liver TG, liver TC, gene and protein expressions involved in lipid metabolism between treated and untreated animals have been statistically analyzed.

Reviewer 2 Report

Comments to authors:

This study investigated the phenolic composition, antioxidant activity and hypolipidemic effect of longan byproducts. It is interesting in its content and valuable to the journals’ reader. Nevertheless, the work needs revision and some issues should be considered before the work could be considered for publication. The followings are some comments and suggestions for authors to consider and improve the manuscript.

1. The extraction rate of longan byproducts polyphenol extract should be provided.

2. Please add some necessary information for the UPLC-QqQ-MS, in particular MRM information. In my opinion, the most important thing for a given scientific report or article should be the repeatability.

3. Why use the concentration of 0.2% LPPE?

4. Line 89, the discussion should be moved to discussion section.

5. Line 89-91, the author reported that corilagin is one of the main phenolic compounds in longan fruits was not be identified in this study due to the unavailable of the standard, however, the authors could identified it by UPLC-QqQ-MS/MS through ionic fragment and expressed it as relative content by quantitative analysis.

6. The detail methods of total polyphenols and antioxidant activities in vitro should be provided.

7. Line 356, the authors reported that LPPE was rich in phenolic compounds and high in antioxidant activity, the comparison with other similar study should be detail discussed.

8. More discussions are required for the hypolipidemic effect observed. What compounds are the main contributors for the effect?

9. There are some grammatical errors in the manuscript. It is advisable for the authors to check throughout the manuscript.

Author Response

This study investigated the phenolic composition, antioxidant activity and hypolipidemic effect of longan byproducts. It is interesting in its content and valuable to the journals’ reader. Nevertheless, the work needs revision and some issues should be considered before the work could be considered for publication. The followings are some comments and suggestions for authors to consider and improve the manuscript.

We appreciate reviewer’s comments, and the manuscript has been revised accordingly.

  1. The extraction rate of longan byproducts polyphenol extract should be provided.

Answer: the extraction rate of longan byproducts polyphenol extract was 10.69%, and this data has been added in the revised manuscript.

  1. Please add some necessary information for the UPLC-QqQ-MS, in particular MRM information. In my opinion, the most important thing for a given scientific report or article should be the repeatability.

Answer: yes, we agree with the reviewer’s opinion. The MRM information including retention time, fragmentor voltages, MS [M-H], and MS/MS data were given in Table 1.

  1. Why use the concentration of 0.2% LPPE?

Answer: in the literatures, 0.1%, 0.2%, 0.5% and even higher concentration of plant polyphenol extracts were usually used. According to our preliminary experiment, 0.2% was relatively low and effective dose.

  1. Line 89, the discussion should be moved to discussion section.

Answer: thanks, those sentences have been moved to the discussion section.

  1. Line 89-91, the author reported that corilagin is one of the main phenolic compounds in longan fruits was not be identified in this study due to the unavailable of the standard, however, the authors could identified it by UPLC-QqQ-MS/MS through ionic fragment and expressed it as relative content by quantitative analysis.

Answer: thanks for this comment. The UPLC-QqQ-MS/MS data was re-analyzed. However, no obvious MS and MS/MS data of corilagin as that reported in the literature was found. Some literatures reported corilagin was one of the main phenolic compounds in longan fruits. However, this compound was not detected in other researches (J. Agric. Food Chem. 2007, 55, 14, 5864–5868; Int. J. Food Prop. 2018, 21, 1, 746-759; Sep. Purif. Technol. 2009, 70,2, 219-224). This difference may be result from the different cultivar, cultivation environment, purification method, and so on. This information was added in the discussion of the revised manuscript.

  1. The detail methods of total polyphenols and antioxidant activities in vitro should be provided.

Answer: thanks for this comment, the detail methods were provided in the revised manuscript.

  1. Line 356, the authors reported that LPPE was rich in phenolic compounds and high in antioxidant activity, the comparison with other similar study should be detail discussed.

Answer: thanks for this comment, the detail discussion was added in our revised manuscript.

  1. More discussions are required for the hypolipidemic effect observed. What compounds are the main contributors for the effect?

Answer: thanks for this comment, more discussions were added. It is speculated that the main components in the LPPE are the main contributors for the effect. In addition, it is necessary to consider the synergistic effects of all the components.

  1. There are some grammatical errors in the manuscript. It is advisable for the authors to check throughout the manuscript.

Answer: thanks for this comment. To make sure the manuscript is properly formatted, we have made a substantial revision in grammar and words throughout text.

Reviewer 3 Report

In general, the article should be improved.

The section on materials and methods is incomplete. It is necessary to indicate in detail the working conditions, the equipment used (brand, model...), etc.

The results section should be improved. The results obtained should be compared with bibliography of similar biomasses and reference everything that the author comments in the manuscript. 

The author has submitted supplementary material that is not commented in the manuscript. In addition, the file does not conform to journal guidelines (review the instructions for authors).

Section 3. Discussion should be improved, it is incomplete.

All of the above comments are detailed in the attached manuscript. I have underlined each part that I think the author should revise and made a comment.

Author Response

In general, the article should be improved.

The section on materials and methods is incomplete. It is necessary to indicate in detail the working conditions, the equipment used (brand, model...), etc.

Answer: the detail information for the materials and methods were provided in our revised manuscript.

The results section should be improved. The results obtained should be compared with bibliography of similar biomasses and reference everything that the author comments in the manuscript.

Answer: thanks for this comment, this section has been improved. We compared our results with those similar reports in the literature and more discussions have been added.

The author has submitted supplementary material that is not commented in the manuscript. In addition, the file does not conform to journal guidelines (review the instructions for authors).

Answer: the supplementary material included the original western images and original data of densitometry readings for all the bands which were required by the editor. This information was edited as Figure 5, and it was commented in our manuscript.

Section 3. Discussion should be improved, it is incomplete.

Answer: we have tried our best to improve the discussion part.

All of the above comments are detailed in the attached manuscript. I have underlined each part that I think the author should revise and made a comment.

Answer: we appreciate reviewer’s patience and carefulness. All the comments marked by the reviewer have been revised accordingly.

Round 2

Reviewer 2 Report

The authors well revised the manuscript and I have no further comments.

Reviewer 3 Report

The comments have been addressed by the author of the manuscript correctly. 

For this reason, my decision is to accept this manuscript for publication.